# Metabonomics and Transcriptomics Analyses Reveal the Development Process of the Auditory System in the Embryonic Development Period of the Small Yellow Croaker under Background Noise

**DOI:** 10.3390/ijms25041954

**Published:** 2024-02-06

**Authors:** Qinghua Jiang, Xiao Liang, Ting Ye, Yu Zhang, Bao Lou

**Affiliations:** State Key Laboratory for Managing Biotic and Chemical Threats to the Quality and Safety of Agro-Products, Institute of Hydrobiology, Zhejiang Academy of Agricultural Sciences, Hangzhou 310000, China; jqh881130@163.com (Q.J.); liangxiao1225@yeah.net (X.L.); 15tye@stu.edu.cn (T.Y.); zhangy@zaas.ac.cn (Y.Z.)

**Keywords:** *Larimichthys polyactis*, auditory system, embryonic development, background noise, metabonome, transcriptome

## Abstract

Underwater noise pollution has become a potential threat to aquatic animals in the natural environment. The main causes of such pollution are frequent human activities creating underwater environmental noise, including commercial shipping, offshore energy platforms, scientific exploration activities, etc. However, in aquaculture environments, underwater noise pollution has also become an unavoidable problem due to background noise created by aquaculture equipment. Some research has shown that certain fish show adaptability to noise over a period of time. This could be due to fish’s special auditory organ, i.e., their “inner ear”; meanwhile, otoliths and sensory hair cells are the important components of the inner ear and are also essential for the function of the auditory system. Recently, research in respect of underwater noise pollution has mainly focused on adult fish, and there is a lack of the research on the effects of underwater noise pollution on the development process of the auditory system in the embryonic development period. Thus, in this study, we collected embryo–larval samples of the small yellow croaker (*Larimichthys polyactis*) in four important stages of otic vesicle development through artificial breeding. Then, we used metabonomics and transcriptomics analyses to reveal the development process of the auditory system in the embryonic development period under background noise (indoor and underwater environment sound). Finally, we identified 4026 differentially expressed genes (DEGs) and 672 differential metabolites (DMs), including 37 DEGs associated with the auditory system, and many differences mainly existed in the neurula stage (20 h of post-fertilization/20 HPF). We also inferred the regulatory mode and process of some important DEGs (*Dnmt1*, *CPS1*, and endothelin-1) in the early development of the auditory system. In conclusion, we suggest that the auditory system development of *L. polyactis* begins at least in the neurula stage or earlier; the other three stages (tail bud stage, caudal fin fold stage, and heart pulsation stage, 28–35 HPF) mark the rapid development period. We speculate that the effect of underwater noise pollution on the embryo–larval stage probably begins even earlier.

## 1. Introduction

The problem of underwater noise pollution has become increasingly serious. Unlike in air, sound travels faster in water and attenuates more slowly [1,2,3], so underwater noise pollution continuously affects all aquatic animals over a long period. Research has shown that frequent human activities are the main cause of underwater environmental noise, including commercial shipping, offshore energy platforms, scientific exploration activities [4,5,6,7], and so forth. Many studies have revealed that underwater environmental noise affects the growth performance, physiological responses, and behaviors of fish [8,9,10,11,12,13,14,15]. Underwater noise pollution has become a potential threat to aquatic animals in the natural environment, and it has also appeared in the aquaculture environment. Intensive culture noise (149 dB re 1 μPa) significantly negatively affects the growth of rainbow trout (*Oncorhynchus mykiss*) [16]. In addition, the larval Atlantic cod (*Gadus morhua*) exhibits lower growth rates and greater yolk sac consumption when disturbed by intensive culture noise for two days [17]. Recirculating aquaculture systems (RASs) are most frequently used in aquaculture, but their noise reduces the welfare of largemouth bass (*Micropterus salmoides*) from the perspectives of growth, physiology, and schooling ability [18]. At present, underwater noise pollution has become an unavoidable problem in the aquaculture industry.

However, some fish show adaptability to noise over a period of time [16,17]. This is because the fish auditory system is different from that of terrestrial vertebrates. Fish have evolved a variety of inner ears and accessory hearing structures (AHSs), and the main difference is the size of the otolith end organs, the shape and orientation of the sensory epithelia, and the orientation patterns of the ciliary bundles of sensory hair cells [19]. In most fish’s inner ears, the otolith comprises small, biomineralized earstones and aids in gravity reception and balance [20]. Otoliths can also be used as biochronometers to archive age and growth information that is essential for evaluating fishery resources and predicting their responses to climate change [21]. Researchers have found that fish otoliths act as a mass load that increases the sensitivity of mechanosensory hair cells to the effects of gravity and linear acceleration [22]. When a sound wave reaches the ears, it generates relative motions between the sensory epithelium and otolith because their significant density difference results in different inertial values; sensory transduction occurs when the relative motions deflect ciliary bundles of hair cells, which allows for fish to hear sound underwater [19,23,24]. However, unlike in humans, where the degeneration or death of mechanosensory hair cells in the inner ear leads to hearing loss and balance disorders, fish sensory hair cells can reproduce after being damaged [25,26]. Therefore, when there is minor damage to the inner ear due to underwater noise, adult fish show some adaptability to the noise over a period of time. It is clear that otoliths and sensory hair cells are the important components of the inner ear in fish auditory systems, and they are also essential for the function of the auditory system in fish.

In recent studies, the focus of the research in respect of underwater noise pollution has mainly been adult fish in the natural environment and the aquaculture environment, and there is a lack of the research on the effects of underwater noise pollution on the early development process of the auditory system in the embryonic development period. Especially in the aquaculture industry, environmental noise pollution persists throughout the development of fish [18]. Further, embryogenesis and embryonic development are the most important life stages in fish, determining fish’s healthy growth and directly affecting the quality and health of future larvae [27]. Whether an embryo–larval fish has the same capacity or potential as an adult fish to adapt to the noise of the culturing environment is still unknown. The small yellow croaker (*Larimichthys polyactis*) is a demersal fish species of the family Sciaenidae and is widely distributed in the Bohai Sea, East Sea, and Yellow Sea of China [28]. *L. polyactis* has high economic and nutritional value, and although artificial aquaculture has been achieved, they are very sensitive to environmental noise because of their keen sense of hearing. When exposed to underwater noise for a long time, they exhibit the obvious avoidance behavior, lose the ability to maintain balance, and may die in severe cases [29]. Therefore, background noise pollution is an urgent problem to be solved in relation to the artificial aquaculture of *L. polyactis*. In early studies, Zhan et al. [30] first discovered and determined the embryonic development schedule of *L. polyactis* via an embryo development observation experiment. They suggested that the otocyst structure appeared in the caudal fin fold stage, and the otoliths could be clearly observed during the heart pulsation stage; this finding provided an important clue for studies on the development of the early auditory system in the small yellow croaker. Before the start of this study, we undertook artificial breeding of *L. polyactis* to obtain embryos in an indoor artificial breeding house. Based on the artificial hatching method, to finish the embryogenesis process, the embryonic development process was also completed indoors. Although we tried to avoid many environmental stimuli (hard light, anoxia, microbial disease, and other unpredictable factors in the natural environment), an uncontrollable aquaculture background noise still existed. For the above reasons, in this study, we selected four important stages of otic vesicle development (the neurula stage, tail bud stage, caudal fin fold stage, and heart pulsation stage) to collect embryo–larval samples, then applied RNA-seq to obtain transcriptomic data and used LC-MS/MS to obtain metabonomic data. We then performed correlation analysis to reveal the development process of the auditory system in the embryonic development period of *L. polyactis* under an aquaculture environment, and to discover the genes/metabolites associated with the auditory system. We finally attempted to determine the initial period of auditory system development and evaluate the effects of aquaculture background noise on the development of the auditory system in the embryo–larval stage.

## 2. Results

### 2.1. The Results of Embryonic Staging and Background Noise Detection

In this study, embryo–larval samples were collected at different times post-fertilization (hours post-fertilization, HPF) [30], including four potentially important stages of otic vesicle development (Figure 1): (1) neurula stage (20 HPF); (2) tail bud stage (28 HPF); (3) caudal fin fold stage (29.5 HPF); (4) heart pulsation stage (35 HPF). Newly hatched fish larvae appeared at 43 HPF, and otoliths could be clearly observed via a dissecting microscope (SAIKEDIGITAL, Shenzhen, China). During the background noise detection, although we used spectral analysis to obtain a sound pressure level (SPL) of 1–24,000 Hz, fish’s perception of the voice frequency range is limited. Therefore, we referred to the range of perceived acoustic frequencies of similar species (*Larimichthys crocea*; its auditory threshold is about 93–99 dB over 100–1000 Hz), and then selected the acoustic wave range of 100–1000 Hz to analyze the SPL [31]. The results of background noise detection (sound of indoor artificial breeding house and underwater sound) are shown in Figure 2. The SPL of the indoor environment sound over 24 h was 29–41 dB (20 µPa), and the SPL of the underwater environment sound (100–1000 Hz) over 24 h was 13.03–78.85 dB (1 µPa).

### 2.2. Analysis of Transcriptomic Data

A total of twelve libraries generated 81.44 Gb of clean data; every library contained at least 6.28 Gb of clean data and the Q30 value was over 93.35% (Table 1). In total, 46,185 assembled transcripts were obtained by mapping to the *L. polyactis* reference genome [32], and all genes were matched with the Gene Ontology (GO) database, Kyoto Encyclopedia of Genes and Genomes (KEGG), Clusters of Orthologous Groups of proteins (COG), Non-Redundant Protein Sequence Database (NR), and Swiss-Prot and Pfam databases information. In total, 27,526 genes were identified, including 18,788 known genes and 8738 new genes. All the raw sequence data were deposited in the National Center for Biotechnology Information (NCBI) Sequence Read Archive (SRA) under BioProject accession number PRJNA1012845.

As shown in Figure 3A, the results of the correlation analysis between biological replicate samples show that the data for the neurula stage group (TR_A1) were very different from those of the other three groups, but that the tail bud stage group (TR_A2), the caudal fin fold stage group (TR_A3), and the heart pulsation stage group (TR_A4) were very similar in terms of sequencing data. We performed differential gene comparisons between the four groups, the results are shown in Figure 3B, and a total of 4026 DEGs (differentially expressed genes) between the four groups were found (Appendix A). Among these DEGs, 725 were upregulated and 1799 were downregulated in the TR_A1_vs_TR_A2 group; 894 were upregulated and 1981 were downregulated in the TR_A1_vs_TR_A3 group; 1059 were upregulated and 2247 were downregulated in the TR_A1_vs_TR_A4 group; 114 were upregulated and 72 were downregulated in the TR_A2_vs_TR_A3 group; 136 were upregulated and 153 were downregulated in the TR_A2_vs_TR_A4 group; and 2 were upregulated and 8 were downregulated in the TR_A3_vs_TR_A4 group. Through the Venn analysis, we found that every pairwise comparison group had the characteristic genes, as shown in Figure 3C. The TR_A1_vs_TR_A2 group had 270 characteristic genes; the TR_A1_vs_TR_A3 group had 272 characteristic genes; the TR_A1_vs_TR_A4 group had 556 characteristic genes; the TR_A2_vs_TR_A3 group had 11 characteristic genes; the TR_A2_vs_TR_A4 group had 29 characteristic genes; the TR_A3_vs_TR_A4 group only had 1 characteristic gene, and evm.TU.Scaffold218.59 was the differential gene common to all pairwise alignment groups. Then, we performed KEGG enrichment analysis of the pairwise comparison groups; the significant enrichment pathway results (padjust < 0.05) are shown in Figure 4. In the TR_A1_vs_TR_A2 group (Figure 4A), the main enrichment pathways were hypertrophic cardiomyopathy (gene number: 45), complement and coagulation cascades (28), protein digestion and absorption (41), dilated cardiomyopathy (46), cardiac muscle contraction (33), ECM–receptor interaction (37), staphylococcus aureus infection (20), ribosome biogenesis in eukaryotes (17), focal adhesion (58) and rheumatoid arthritis (22). In the TR_A1_vs_TR_A3 group (Figure 4B), the main enrichment pathways included ribosome biogenesis in eukaryotes (29), protein digestion and absorption (42), complement and coagulation cascades (27), hypertrophic cardiomyopathy (43), ECM–receptor interaction (40), dilated cardiomyopathy (45), focal adhesion (65), cardiac muscle contraction (31), and arrhythmogenic right ventricular cardiomyopathy (34). ECM–receptor interaction (49), protein digestion and absorption (48), ribosome biogenesis in eukaryotes (27), complement and coagulation cascades (30), hypertrophic cardiomyopathy (48), dilated cardiomyopathy (51), amoebiasis (46), focal adhesion (74), glycolysis/gluconeogenesis (20), cardiac muscle contraction (32), arrhythmogenic right ventricular cardiomyopathy (36), and histidine metabolism (10) were the main enrichment pathways in the TR_A1_vs_TR_A4 group (Figure 4C). In the TR_A2_vs_TR_A3 group (Figure 4D), the main enrichment pathways were necroptosis (10), osteoclast differentiation (9), NF-kappa B signaling pathway (8), complement and coagulation cascades (6), IL-17 signaling pathway (6), influenza A (7), NOD-like receptor signaling pathway (7), and coronavirus disease-COVID-19 (8). The TR_A2_vs_TR_A4 group and the TR_A3_vs_TR_A4 group had no main enrichment pathways (significance assessment criteria: padjust < 0.05). The KEGG enrichment analysis showed that the TR_A1 group had more differences than the other three groups. We also found 37 DEGs associated with the auditory system (Table 2). Finally, we counted and classified the significant enrichment pathways of all DEGs (Appendix A) by using the multiple test correction method (Benjamini–Hochberg, BH). These 37 DEGs were significantly enriched in 15 pathways, including the PI3K-Akt signaling pathway, focal adhesion, ECM–receptor interaction, adrenergic signaling in cardiomyocytes, cardiac muscle contraction, bile secretion, proximal tubule bicarbonate reclamation, and fatty acid biosynthesis.

### 2.3. Analysis of Metabonomic Data

Metabolites were extracted from two groups’ samples; one group (ME_NS) comprised the neurula stage (20 HPF) samples and the other group (ME_HtS) comprised a mix of samples of 28–35 HPF; every group had six replicates. After removing the metabolites with missing values ≥ 50%, a total of 1977 metabolites (881 neg and 1096 pos) were found in twelve samples (Appendix A). Then, we performed partial least squares discriminant analysis (PLS-DA), which made the division of the total variation into two major components (Component 1 and Component 2); these components contributed 69 and 19.2% of the variation, respectively (Figure 5A). 

The differential metabolites (DMs) (VIP > 1, *p* < 0.05) between ME_NS and ME_HtS totaled 672 (363 negative and 309 positive); this result is visualized using volcano plots (Figure 5B,C). Among these DMs, 208 were increased and 155 were decreased in the negative pattern group (Figure 5B), and 166 were decreased and 143 were decreased in the positive pattern group (Figure 5C). The cluster and heatmap analysis results for DMs (top 50) are shown in Figure 5D. Among the ten subclusters (left side in figure), 18 metabolites were increased and 32 metabolites were decreased between ME_NS and ME_HtS. Then, we performed KEGG enrichment analysis of the DMs (363 negative and 309 positive). The results show the top 20 significant enrichment pathways (Figure 6A), including adrenergic signaling in cardiomyocytes (metabolite number: 1); one carbon pool by folate (1); phototransduction (1); glycosylphosphatidylinositol (GPI)-anchor biosynthesis (1); autophagy-animal (1); PPAR signaling pathway (1); autophagy—other (1); pantothenate and CoA biosynthesis (3); ferroptosis (3); sphingolipid metabolism (3); necroptosis (2); citrate cycle (TCA cycle) (3); GnRH signaling pathway (2); FoxO signaling pathway (2); linoleic acid metabolism (4); mTOR signaling pathway (2); alanine, aspartate, and glutamate metabolism (5); arachidonic acid metabolism (8); purine metabolism (10), and nucleotide metabolism (12). Next, all the DMs were mapped to the HMDB to complete the classification of chemical compounds; the results are shown in Figure 6B. Among these metabolites, the main confirmed chemical compounds included amino acids, peptides, and analogues (143); fatty acids and conjugates (22); carbohydrates and carbohydrate conjugates (20); and others. We also used the BH method to count and classify the significant enrichment pathways of DMs (Appendix A); most pathways were related to metabolism (96), environmental information processing (13), cellular processes (5), genetic information processing (5), and organismal systems (3).

### 2.4. Correlation Analysis between the Transcriptomic and Metabonomic Data

In order to better reveal the regulation changes in vivo in the embryo–larval stage in an aquaculture environment, and to study the effects of aquaculture environment noise on embryo–larval development, we performed a correlation analysis between the transcriptome and metabonome. First, we examined the joint KEGG pathway enrichment analysis between the transcriptome and metabonome to count and compare the identical enrichment pathways. We found eleven identical enrichment pathways between genes and metabolites (Figure 7A); these were glycolysis/gluconeogenesis; nucleotide metabolism; purine metabolism; pyrimidine metabolism; linoleic acid metabolism; alanine, aspartate, and glutamate metabolism; ABC transporters; arachidonic acid metabolism; biosynthesis of cofactors; FoxO signaling pathway; and mTOR signaling pathway. Next, we evaluated the intrinsic correlation between the transcriptomic and metabonomic data by using two-way orthogonal partial least squares (O2PLS), which enabled us to calculate the score of each sample and obtain a joint score (*pq* value), where “*p*” represents the load value of the gene and “*q*” represents the load value of the metabolite. We were then able to obtain a load map (Figure 7B). The metabolites/genes of high loading values were considered necessary for evaluating the similarity and relevance between two datasets. Finally, based on the absolute load values, we selected the top 10 DMs/DEGs to construct a histogram (Figure 7C). In the common annotated pathways between DEGs and DMs (Appendix A), we found that the auditory system-associated DEGs were mainly annotated to six pathways, which were neuroactive ligand–receptor interaction, adrenergic signaling in cardiomyocytes, vascular smooth muscle contraction, FoxO signaling pathway, necroptosis, phosphatidylinositol signaling system, gap junction, GnRH signaling pathway, and nitrogen metabolism.

### 2.5. Confrmation of DEGs via qRT-PCR

To validate the reliability and stability of the RNA sequencing data of the DEGs, we selected six DEGs for qRT-PCR validation, and *β-actin* (MT330378) was used as a housekeeping gene. These six DEGs were *CA2*, endothelin-1, *GRM1*, *GRM7*, *CACNA1S,* and *Dnmt1* (Appendix A). The primer information is shown in Appendix A and the relative expression level results are shown in Figure 8; these results suggest that the transcriptomic data are reliable.

## 3. Discussion

In early studies on the development of the auditory system in *L. polyactis*, researchers found that the otocyst structure appeared in the caudal fin fold stage, and otoliths could be clearly observed in the heart pulsation stage [30], suggesting that rapid development of the auditory system occurred between the tail bud stage and caudal fin fold stage. Thus, we inferred that the initial period of development of the auditory system might occur in the neurula stage, because the neurula stage is an important period in animals’ embryonic development and is the initial stage of most nervous system development. Finally, a total of 37 DEGs (18 downregulated and 19 upregulated) (Table 2) associated with the auditory system were found in our study. Among these DEGs, the main differences existed in the neurula stage group (TR_A1) (Figure 3); these results also verify our previous conjecture. However, the auditory structure of fish is different from that of terrestrial vertebrates. Most fish analyze the movement of their body in the sound field using the otolith in the inner ear [19]. The otolith grows continuously from the embryo stage and is encased within fluid-filled sacs that become the complete organs of the inner ear [21]; the otolith is necessary for auditory perception in most fish species [21,22]. In our study, we found many genes involved in otolith formation, including some that have been previously reported, such as otopetrin-2 (*OTOP2*) [33], α-Tectorin [34], otolin-1 (evm.TU.Scaffold69.333; evm.TU.Scaffold69.334) [22], otogelin (*Otog*) (evm.TU.Scaffold13.238) [22], otolith matrix protein-1 (*OMP-1*) [21], cochlin (evm.TU.Scaffold252.297) [35], and others, but not all of these had differential expression in the four development stages. This suggests that there are essential and non-essential genes in the early stage of otolith formation, and that the essential genes for otolith formation might also be different in different animals. Among the identified DEGs, 15 were significantly upregulated in the neurula stage. These DEGs are likely to be important participants in the early development of the auditory system in the neurula stage; many have been extensively studied in vertebrates, but some have not frequently been mentioned in fish.

The formation process of otoliths includes occurring and calcification; first, the otolith precursor particles bind to the tips of the immotile kinocilia of tether cells via the *Otog* protein, which completes the otolith nucleation occurring [22]. However, in this study, we found that *Otog* showed no significant changes from the neurula stage to the heart pulsation stage, probably because the *Otog* protein is a protein only required for otolith tethering in the otolithic membrane but is not the main protein for daily increments of otolith formation. In this process, many integrins have been found to be involved in the polymerization process of otolith precursor particles [36]; for example, integrin α-3 and α-6 are highly expressed in the mouse inner-ear region [37]; integrin α8β1 regulates hair cell differentiation and stereocilia maturation in the mouse ear [38] and can also bind osteopontin, which is a component of rodent otoconia and is similar to otoliths in fish [39]. Similarly, integrin α-3, α-5, α-6, and β-1 significantly change in the neurula stage, suggesting they are important regulators in the polymerization process of otolith precursors in *L. polyactis*.

After nucleation, the second process is calcification, and this requires sufficient calcium ions. In early reports, the NADPH oxidase (NOX) family and NOX complex are considered the primary participators in the process [39]. In this study, we found that *NOX1* and *NOX4* were significantly highly expressed in the neurula stage, and two NOX complex functional genes (neutrophil cytosol factor 2 (*ncf2*) and NADPH oxidase organizer 1 (*Noxo1*)) [40,41] were also significantly highly expressed in the same stage. This suggests that *NOX1*, *NOX4*, *ncf2*, and *Noxo1* are probably important regulators of the otolith calcification process in *L. polyactis*. Furthermore, some researchers have found that *OTOP1* and *OTOP2* function as proton channels and respond to ATP in the endolymph to increase intracellular calcium levels during otolith mineralization in mice and zebra fish [42,43]. Interestingly, we only found *OTOP2* in the transcriptome database of this study, and this was significantly highly expressed in the neurula stage; this might imply that *OTOP2* is the key regulator of the otolith calcification process in the neurula stage of *L. polyactis*. In order to maintain the growth and calcification of the otolith, there is a need to maintain pH stability and enough bicarbonate, and the carbonic anhydrase (CA) family comprises the important participants in this work. In the inner ear of *Oncorhynchus mykiss*, carbonic anhydrase isozymes act to produce bicarbonate ions from carbon dioxide in the epithelial cells and then supply these to the endolymph [44]. In mice, *CA2* and *CA6* are likely to cooperate with V-ATPase in the plasma membrane of epithelial cells to maintain pH [45]. In our study, we found that only *CA2*, *CA4,* and *CA6* had significantly differential expression; among them, *CA2* was significantly highly expressed in the neurula stage, and *CA4* and CA6 showed sustained high expression in the middle and late developmental stages (28–35 HPF). This result suggests that *CA2*, *CA4,* and *CA6* are important participants in otolith calcification and play their respective roles in different development periods. In addition, to maintain the growth of the otolith, a major incremental protein for daily increments is needed. In *Oncorhynchus mykiss*, *OMP-1* and otolin-1 are the probable candidate proteins for daily increment formation, especially otolin-1 [46]. However, we found that only *OMP-1* showed high expression in the middle and late developmental stages (28–35 HPF) and otolin-1 (evm.TU.Scaffold69.333) had no significant change; this suggests that *OMP-1* is the most probable candidate protein for the daily increment of otolith formation in *L. polyactis*.

In addition to otolith development, the development of the fish auditory system also includes the development of the inner ear organ and auditory conduction system. Mice and zebra fish have been used as model animals for human auditory disease research, which has greatly facilitated the study of the mechanisms and genes involved in auditory development and disease. In zebra fish, endothelin-1 has been implicated in inner ear development and function [22,47], and endothelin-1 mRNA is turned on during the critical period of otolith nucleation [48]; this can also be detected in the otic vesicle at 24 HPF [49]. endothelin-1 and its receptors are both enriched in the inner ear support cells of adult zebra fish [22,50]. Similarly, in our study, endothelin-1 was significantly highly expressed in the neurula stage (20 HPF), which also implies that endothelin-1 has an important role in the early development of the inner ear. Furthermore, the absence of FMRP (fragile X mental retardation protein) induces fragile X syndrome and also induces modest peripheral hearing loss in mice and humans [51]. In this study, we found that FMRP (evm.TU.Scaffold630.3) exhibited no significant change in the four development stages, but we did find another FMRP interactor, *NUFIP1* (nuclear FMRP-interacting protein 1). This is an RNA-binding protein with an expression profile matching that of FMRP, which was significantly highly expressed in the neurula stage. According to previous research, in mice, *NUFIP1* is involved in the export and localization of mRNA and interacts with FMRP to regulate local protein synthesis near synapses; it is also involved in the neuronal maturation process [52]. These results likely suggest that the inner ear development of *L. polyactis* is healthy in the neurula stage. In a recent study, researchers found that *Lmx1a* [53] and *Pax2* [54] are essential genes for normal development of the inner ear in mice; the results of their separate gene knockdown experiment showed severe cochlear defects as well as the absence of the spiral ganglion. These two genes had a high expression level at 20–29.5 HPF (neurula stage to caudal fin fold stage). This period marks the stage of inner ear formation and rapid development; thus, we speculate that *Lmx1a* and *Pax2* also have important functions in the early development of the inner ear in *L. polyactis*.

The auditory conduction system is an important segment of the auditory system. For example, the excitatory synapses cover the region from the cochlear hair cells to the auditory cortex in the auditory pathway, and they utilize L-glutamate as the primary neurotransmitter to transmit all kinds of stimulus information [55]; this includes acoustic information. Therefore, the glutamate receptor is the important conduction structure between synapses. In our transcriptome database, we found many glutamate receptor genes, such as *GRM1* (metabotropic glutamate receptor-1), *GRM7*, *GRIN1* (glutamate receptor ionotropic NMDA 1) (evm.TU.Scaffold67.84), and so on, but only *GRM1* was significantly upregulated in the neurula stage. Curry et al. [56] found that GRM1 modulation of glycinergic frequency persists throughout development and may persist in the mature auditory system, and they suggested that the spontaneous glycinergic transmission in the MNTB (medial nucleus of the trapezoid body; the MNTB is a critical nucleus in the auditory brainstem nuclei involved in sound localization) is facilitated by *GRM1*; this conclusion has also been recognized and supported by other researchers [57]. Based on the above reports and our study, we speculate that *GRM1* is probably an important indicator gene for the functional development of the auditory system. We also found other important function genes of the auditory conduction system: *USH2A*, whose mutations can induce a disorder characterized by retinitis pigmentosa and mild-to-severe hearing loss [58,59]; neuropilin-2, which is an important axon-guidance receptor in spiral ganglion neurons (SGNs), and SGNs are bipolar afferents linking the peripheral and central auditory systems—their mutation and loss induce congenital or noise-induced hearing loss [60]; *Cntnap2*, one of the autism-linked genes, which participates in the regulation of the auditory processing pathway—its knock-out induces the response delay of the auditory brainstem response [61]; ephrin type-A receptor, an Eph receptor ligand that mediates diverse functions at multiple levels of the auditory pathway, including axon guidance and targeting, and cell migration [62]; transmembrane protease serine [63]—mutations of its coding gene (*tmprss3*) can cause non-syndromic deafness. All of these important functional genes were significantly highly expressed in the neurula stage of *L. polyactis*, suggesting that the early development time of the auditory system necessarily begins at the neurula stage at latest.

There were some other downregulated DEGs associated with the auditory system that were found in neurula stage; they all showed sustained upregulation from the neurula stage to the heart pulsation stage. We infer that these 18 downregulated DEGs play an important role in the maturation and functionalization process of the auditory system, and also have respective features in different development periods, such as those mentioned above; for the CA family members *CA2*, *CA4* and *CA6*, their function in regulating and maintaining pH might have the characteristics of time difference. In this study, we also found many genes of neuroreceptors, transporters, channel proteins, structural constituents, and enzymes, all of which showed sustained high expression until the middle and late developmental stages. Among them, most have been found to play important roles in the function of the auditory system, including GABA A receptor (gamma-aminobutyric acid receptor subunit β-3) [64,65], *GRM7* [55,66], *CACNA1S* (voltage-dependent L-type calcium channel subunit α-1S) [3], *SLC1A1* (excitatory amino acid transporter 3) [3], *TRPM2* (transient receptor potential cation channel subfamily M member 2) [67], ephrin type-B receptor 1-B [62], collagen α-1(XI) chain (*col11a1*) (involved in the sensory perception of sound) [3,68], fatty acid synthase (polyketide synthase dehydratase activity) [36], *SPTB* (spectrin β chain, non-erythrocytic 1) (cell signaling function) [69], osteonectin (the structural constituents of the otolith) [21,70], neuroserpin (the structural constituents of the otolith) [21,70], *Six1* (an important gene for the morphogenesis of the cochlea and the posterior ampulla in mice) [71], and *Dnmt1* (DNA methyltransferase 1) (an essential gene for regulating the development of auditory organs in zebrafish) [72].

In our study, a total of 1977 metabolites were found. Among them, there were 672 DMs (363 negative and 309 positive) (Appendix A), and the results of the KEGG enrichment analysis show that the main differences were the important pathways of the development process (Figure 5A). For better revealing and understanding the early development of the auditory system in the embryo–larval stage, and to research the regulation function of the DEGs in metabolism pathways, we performed a correlation analysis between DEGs and DMs. Finally, we found 66 common annotated pathways between DEGs and DMs (Appendix A). Among these pathways, eight pathways were commonly annotated with 9 auditory-relevant DEGs and 11 DMs (Table 3), including nitrogen metabolism (lco00910), vascular smooth muscle contraction (lco04270), neuroactive ligand–receptor interaction (lco04080), GnRH signaling pathway (lco04912), Gap junction (lco04540), FoxO signaling pathway (lco04068), cysteine and methionine metabolism (lco00270), and adrenergic signaling in cardiomyocytes (lco04261). Through the KEGG pathway analysis, we examined the interaction between DEGs and DMs and tried to find the potentially important metabolites associated with the development of the auditory system.

In cysteine and methionine metabolism, *Dnmt1* is an important regulator of S-Adenosylhomocysteine synthesis, and its downregulation affects L-homocysteine metabolism. According to one report, the downregulation of *Dnmt1* leads to malformed otoliths and deformed semicircular otoliths [72]. Furthermore, homocysteine can affect the tissue mineral density of cortical bone, osteocyte reprogramming, apoptosis, and mineralization [73]; this process is similar to the calcification process of otoliths [21]. We also found that S-adenosylhomocysteine (pos_1457) was downregulated in the neurula stage, which was the same as *Dnmt1*; so, we suggest that *Dnmt1* might be regulating S-adenosylhomocysteine metabolism to affect the L-homocysteine expression level, then affect or regulate the development of otoliths in *L. polyactis*. CA family enzymes are important participants in offering bicarbonate and pH maintenance [44,45] and nitrogen metabolism. CA family enzymes convert carbon dioxide into hydrogen carbonate; hydrogen carbonate and ammonium ions are converted to carbamoyl phosphate (carbamoyl-P) by *CPS1* (carbamoyl-phosphate synthase); and carbamoyl-P is one of the urea cycle (UC) intermediates [74]. In addition, the conversion process of carbamoyl-P releases carbon dioxide. In this series of cyclic reactions, *CPS1* might function as a rate-limiting enzyme. Interestingly, *CPS1* is a DEG (evm.TU.Scaffold254.439) that was downregulated in the neurula stage, suggesting that *CPS1* affects the metabolic process of hydrogen carbonate and might be an upstream effect gene of the CA family enzymes in offering bicarbonate and pH maintenance. Moreover, endothelin-1 plays an important role in inner ear development [22,47,48,49,50]. In neuroactive ligand–receptor interaction, endothelin-1 activates *EDNRB* (endothelin receptor type B) and *EDNRB* activates multiple downstream function genes, including *PLA2G* (group XIIA secretory phospholipase A2) and *PLA2G4* (cytosolic phospholipase A2 zeta), which are important regulators in arachidonic acid metabolism, finally inducing the production of 20-HETE (20-hydroxyeicosatetraenoic acid) [75]. In addition, 20-HETE can activate the NOX system to increase superoxide production [76]. Coincidentally, the NOX system (NOX family and NOX complex) is the primary participator in the otolith calcification process [39]. Furthermore, *EDNRB* (evm.TU.Scaffold588.321), *PLA2G* (evm.TU.Scaffold1931.186), and *PLA2G4* (evm.TU.Scaffold282.6) also exhibited significant differential expression in this study, and arachidonic acid (neg_4165) was also downregulated. Thus, we infer that endothelin-1 might influence the development process of the inner ear via *EDNRB* activating arachidonic acid metabolism to regulate the NOX system. Moreover, in neuroactive ligand–receptor interaction, *GRM1*, *GRM7*, GABA A receptor, and *CACNA1S* are the important receptors of neurons, and are also important signal receivers of the perception system [3,55,56,57,66]. Further, through our correlation analysis, we found that glutamate, GABA, and calcium ions are the primary identification signals, respectively.

## 4. Materials and Methods

### 4.1. Animals

The wild *L. polyactis* were caught in the East Sea of China, then cultured in marine aquaculture cages (Xiangshan County, Ningbo, China). In 2014, the first artificial breeding experiment was completed in the Xiangshan harbor aquatic seeding Co., Ltd. (Xiangshan County, Ningbo, China). The small yellow croaker develops to a sexually mature stage in one year, so all fish were aquacultured for one year at least. Then, we selected some fish to transfer into 40 m^3^ ponds (water temperature: 14–16 °C) in an indoor artificial breeding house to complete the artificial mating from March to April every year. The larvae were fed with rotifers (*Brachionus* sp.) initially, and then microalgae (*Chlorella* sp.) in 1.6 m^3^ (1600 L) breeding buckets for 15 days. For another 15 days, the larvae were fed with newly hatched *Artemia nauplii* (Aquamaster, Binzhou, China) until developing to the juvenile stage, and then fed with a standard commercial diet. After one month, these juvenile fish were transferred to the marine aquaculture cages for culturing. So far, we have successfully cultivated eight generations. In this study, we referred to the research report of Zhan et al. [30], which suggested that the caudal fin fold stage and the heart pulsation stage are the important periods of auditory system development in the small yellow croaker. In addition, the neurula stage is an important stage in the embryonic development of fish. Therefore, according to the different times of post-fertilization (hour of post-fertilization, HPF) and following the chronological order of embryonic development [30], we then selected four potentially important stages of otic vesicle development (the neurula stage, tail bud stage, caudal fin fold stage, and heart pulsation stage) to collect embryo–larval samples. First, we used 10 parent fish (5 male and 5 female, body weights of approximately 60–70 g, body length range of 14–15 cm) to complete the artificial mating, then transferred the fertilized eggs to the 1.6 m^3^ (1600 L) breeding bucket and opened the underwater oxygen supply valve to hatch them (water temperature: 14–16 °C). At intervals of 30 min to 1 h, we used a dissecting microscope (SAIKEDIGITAL, Shenzhen, China) to observe and classify the different development stages of embryonic larvae, which were bred in a 1.6 m^3^ (1600 L) breeding bucket; then, we collected the embryo–larval samples into 1.5 mL EP tubes by tweezers, with at least 20 embryonic larvae (egg diameter: 1100–1300 µm) being obtained in every tube (50–80 mg per tube). In total, approximately 50 tubes were collected and stored at −80 °C. The above sample collection work was completed in the Xiangshan harbor aquatic seeding Co., Ltd. (Xiangshan County, Ningbo, China) in April 2023.

### 4.2. The Environment Sound Detection

In this study, the environment sound mainly included the sound of the indoor artificial breeding house and underwater sound in the 1.6 m^3^ breeding bucket (open oxygen supply equipment). The sound detection of the indoor environment was assessed using a sound level meter (Aihua, Hangzhou, China). The underwater environment sound was investigated using an HTD42 hydrophone (TD, Beijing, China) and a sound collector (TD, Beijing, China) to complete collection, and the Server_Monitor v1.0.4 software (TD, Beijing, China) was used for raw reads preprocessing. Then, we used WavConverter v1.0 software (TD, Beijing, China) to analyze the collected data and converted these to audio data. Finally, we used Audacity v3.4.2 software (https://www.audacityteam.org/ (accessed on 11 April 2023)) to finish the spectrum analysis. This work was finished in April 2023.

### 4.3. Total RNA Isolation and Illumina Sequencing

A total of 12 tubes (50 mg per tube) of samples collected in April (2023) were used for RNA-seq, and every development stage had three tubes. The total RNA was isolated using QIAzol Lysis Reagent (Qiagen, Dusseldorf, Germany) following the manufacturer’ s procedure. The total RNA quantity and purity were confirmed using a NanoDrop 2000 Spectrophotometer (Thermo Fisher Scientific, Cheshire, UK), followed by use of agarose gel electrophoresis to detected the RNA integrity. Lastly, we used an Agilent 5300 (Agilent, Santa Clara, CA, USA) to measure RIN number. For RNA sequencing, the total RNA was subjected to poly (A) enrichment by using oligo (dT) magnetic beads. Approximately 1 ug of total RNA were used to prepare an sRNA library according to the protocol of the Illumina^®^ Stranded mRNA Prep Kit (Illumina, San Diego, CA, USA). Sequencing was conducted using an Illumina Novaseq 6000 platform (Majorbio, Shanghai, China).

### 4.4. Alignment of Transcriptomic Data

Based on the base mass, base error rate, and base content, we analyzed the raw sequences and filtered out the low-quality sequences. The resulting high-quality sequences were mapped onto the *Larimichthys polyactis* reference genome [32] by using TopHat2 v2.1.1 and HISAT2 v2.0.5 [77,78]. Then, we used Cufflinks v0.8.0 software to complete the mapped reads’ assembly and function annotation [79]. The RSEM v1.3.3 software was used to generate the total number of reads for each gene [80]. The analysis of differential gene expression was performed using DESeq2 v1.42.0 software [81], and DEGs were considered statistically significant if they had a FDR (false discovery rate) value ≤ 0.05 and an absolute value of log2 (fold change) ≥ 1.

### 4.5. GO and KEGG Pathway Enrichment Analysis

All the obtained genes were translated to protein sequences, and BLAST in SwissProt by using DIAMOND v2.1.8 (https://github.com/bbuchfink/diamond (accessed on 20 June 2023)) [82]. Next, we completed Clusters of Orthologous Groups of proteins (COG), Gene Ontology (GO), and Kyoto Encyclopedia of Genes and Genomes (KEGG) annotation analyses. The GO term enrichment analysis of every mapped gene was performed by using Goatools v0.7.6 (https://github.com/tanghaibao/GOatools (accessed on 20 June 2023)) [83], and we used the R package clusterProfiler v4.0 to finish the KEGG pathways enrichment analysis [84].

### 4.6. Metabolite Extraction and UPLC-MS/MS Analysis

A total of 24 tubes (50–80 mg per tube) of samples collected in April (2023) were used for metabolite extractions. In this study, we took these samples from two groups: one group was from neurula stage (20 HPF), and the other group was a mix samples of the tail bud stage, caudal fin fold stage, and heart pulsation stage (28–35 HPF). Every group had six parallel samples. The samples, as 50 mg solids, were put into 2 mL centrifuge tubes, and a 6 mm diameter grinding bead was added. Then, 400 μL of extraction solution (methanol/water = 4:1 (*v*/*v*)) containing 0.02 mg/mL of internal standard (L-2-chlorophenylalanine) was used for metabolite extraction. Samples were ground by using the Wonbio-96c (Shanghai wanbo biotechnology Co., Ltd. #32, Xinxin Road, Chedun Town, Songjiang District, Shanghai, China) frozen tissue grinder for 6 min (−10 °C, 50 Hz), followed by low-temperature ultrasonic extraction for 30 min (5 °C, 40 kHz). The samples were left at −20 °C for 30 min, centrifuged for 15 min (4 °C, 13,000× *g*), and then the supernatant was transferred into the injection vial for LC-MS/MS analysis. The LC-MS/MS analysis of sample was conducted using a SCIEX UPLC-Triple TOF 5600 system equipped with an ACQUITY HSS T3 column (100 mm × 2.1 mm i.d., 1.8 μm; Waters Corporation, Milford, MA, USA) at Majorbio Bio-Pharm Technology Co., Ltd. (Shanghai, China). The mobile phases consisted of 0.1% formic acid in water/acetonitrile (95:5, *v*/*v*) (solvent A) and 0.1% formic acid in acetonitrile/isopropanol/water (47.5:47.5, *v*/*v*) (solvent B). The flow rate was 0.40 mL/min and the column temperature was 40 °C.

### 4.7. Metabolites Data Analysis

The metabolite raw data were assessed by using Progenesis QI v2.3 (Waters Corporation, Milford, MA, USA) software, and a three-dimensional data matrix in CSV format was exported. The information in this three-dimensional matrix included sample information, metabolite name, and mass spectral response intensity. Internal standard peaks, as well as any known false positive peaks (including noise, column bleed, and derivatized reagent peaks), were removed from the data matrix, which was deredundant and peak-pooled. At the same time, the metabolites were identified by searching databases, and the main databases were the HMDB (http://www.hmdb.ca/ (accessed on 25 August 2023)), Metlin (https://metlin.scripps.edu/ (accessed on 25 August 2023)), and Majorbio Database. The data were analyzed through the free online platform of the Majorbio cloud platform (cloud.majorbio.com (accessed on 25 August 2023)) [85]. Metabolic features that were detected in at least 80% of any set of samples were retained. After filtering, minimum metabolite values were imputed for specific samples in which the metabolite levels fell below the lower limit of quantitation, and each metabolic feature was normalized by sum. To reduce the errors caused by sample preparation and instrument instability, the response intensity of the sample mass spectrum peaks was normalized by the sum normalization method, and then the normalized data matrix was obtained. Meanwhile, variables with a relative standard deviation (RSD) > 30% of QC samples were removed, and log10 processing was performed to obtain the final data matrix for subsequent analysis. Perform variance analysis on the matrix file was conducted after data preprocessing.

### 4.8. Identification of Differential Metabolites, and Pathway Enrichment Analysis

The R package “ropls” (v1.6.2) was used to perform principal component analysis (PCA), orthogonal least partial squares discriminant analysis (OPLS-DA), and 7-cycle interactive validation evaluating the stability of the model. The metabolites with VIP > 1, *p* < 0.05 were determined to be significantly different metabolites based on the variable importance in the projection (VIP) obtained by the OPLS-DA model and the *p*-value generated by the Student’ s *t*-test. Differential metabolites (DMs) among the two groups were mapped into their biochemical pathways through metabolic enrichment and pathway analysis based on the KEGG database (http://www.genome.jp/kegg/ (accessed on 26 August 2023)). These metabolites could be classified according to the pathways they involved or the functions they performed. Enrichment analysis was used to analyze a group of metabolites in a function node of whether they appeared or not. The principle was that the annotation analysis of a single metabolite develops into an annotation analysis of a group of metabolites. The Python package “scipy.stats” (v1.11.4) (https://docs.scipy.org/doc/scipy/ (accessed on 26 August 2023)) was used to perform enrichment analysis to obtain the most relevant biological pathways for experimental treatments.

### 4.9. Validation of RNA-Seq Data via qRT-PCR

Another 12 tubes (50–80 mg per tube) of samples collected in April (2023) were used for extracting total RNA, and every development stage had three tubes. All samples were ground with liquid nitrogen, the total RNA was isolated by the Trizol method (Vazyme, Nanjing, China), and then we used a NanoDrop 2000 UV Spectrophotometer (Thermo Fisher Scientific, Cheshire, UK) to determine the RNA concentrations. After, we used 10 × gDNA Remove Mix (Takara, Kyoto, Japan) to remove the genomic DNA, then used the Hifair^®^ II 1st Strand cDNA Synthesis Kit (Yeasen, Shanghai, China) according to the manufacturer’ s protocol to synthesize the first strand of cDNA. This cDNA was stored at −80 °C until use. qRT-PCR (quantiative real-time PCR) testing was carried out using the ABI 7500 qPCR instrument (Thermofisher, Sunnyvale, CA, USA) according to the manufacturer’ s instructions of the SYBR^®^ Premix Ex Taq™ II Kit (Takara, Kyoto, Japan). PCR conditions were as follows: 95 °C for 2 min, 40 cycles of 95 °C for 15 s, and 60 °C for 20 s. Additional melting curve analysis was performed to confirm the product specificity, during which the temperature increased from 55 to 95 °C at a rate of 0.2 °C/s. For each tested sample, the reactions were carried out in triplicate for technical replicates. The reference gene was *β-actin*, and the relative mRNA expression levels were calculated using the comparative Ct (2^−∆∆Ct^) method [86]. Statistical significance in this study was analyzed using the SPSS 19.0 software (IBM North America, 590 Madison Avenue New York, USA). All data were subjected to normality testing using the Kolmogorov–Smirnov and Cochran tests prior to all statistical tests. Significant differences were accepted at *p* < 0.05 using one-way ANOVA followed by the Student’ s *t*-test or Tukey test.

## 5. Conclusions

In summary, through transcriptome and metabonome analyses, we revealed the development process of the auditory system in the embryonic development period of the small yellow croaker in background noise (indoor environment sound: 29–41 dB (20 µPa); underwater environment sound: 13.03–78.85 dB (1 µPa)). Of the 27,526 genes discovered, we identified 4026 DEGs, including 37 DEGs associated with the auditory system, and their main differences were noted in the neurula stage. These results also verify our previous conjecture (that the initial period of development of the auditory system probably begins in the neurula stage). Further, a total of 1977 metabolites were found, including 672 DMs. Their KEGG analysis showed that the main differences were in many important pathways of the development process, and these differences mainly existed between the neurula stage (20 HPF) and the other three stages (tail bud stage to heart pulsation stage, 28–35 HPF). In addition, through the correlation analysis between DEGs and DMs, we inferred the regulatory mode and process of some important DEGs during the development of the auditory system. The results suggest that the neurula stage is the important period of otolith formation and inner ear development, and the other three stages mark the rapid development period of the auditory system in *L. polyactis*. Finally, based on these findings and results, we inferred that the auditory system development in *L. polyactis* begins at least in the neurula stage or earlier; this also means that embryo–larval *L. polyactis* are exposed to environmental noise earlier. Although no serious harmful effects were found in this study, this may be due to the collected samples not being sufficient and the lack of an absolutely quiet environment for the control group. However, we still feel that background noise pollution is an important problem in both natural and aquaculture environments that requires attention and solution.

## Figures and Tables

**Figure 1 ijms-25-01954-f001:**
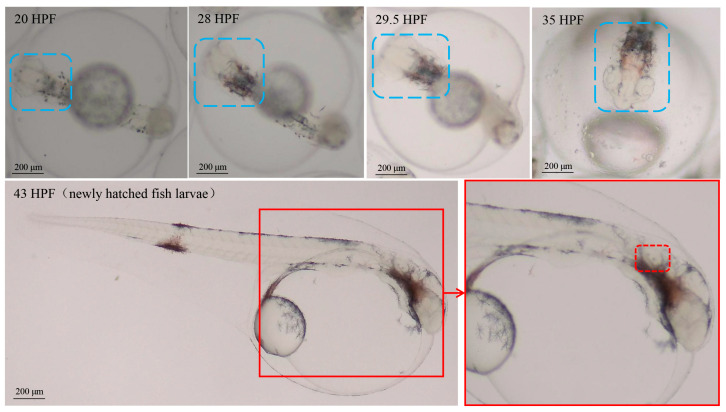
The staging for embryonic development of *Larimichthys polyactis*. The area marked by blue dotted boxes represents the embryo-larval brain in different stages; the area marked by red dotted boxes represents the otoliths of fish larvae.

**Figure 2 ijms-25-01954-f002:**
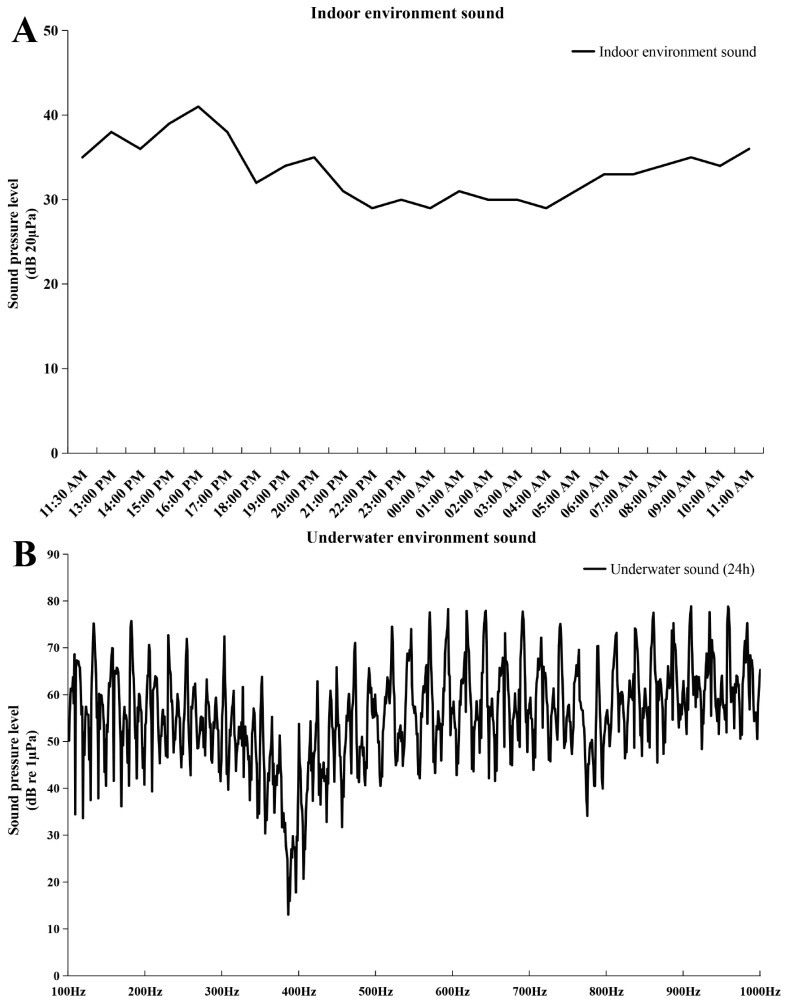
Sound pressure level (SPL) of background noise. (**A**) SPL of indoor environment sound. The abscissa represents the detection time, and the ordinate represents SPL (20 µPa); (**B**) SPL of underwater environment sound. The abscissa represents the frequency (100–1000 Hz), and the ordinate represents SPL (1 µPa).

**Figure 3 ijms-25-01954-f003:**
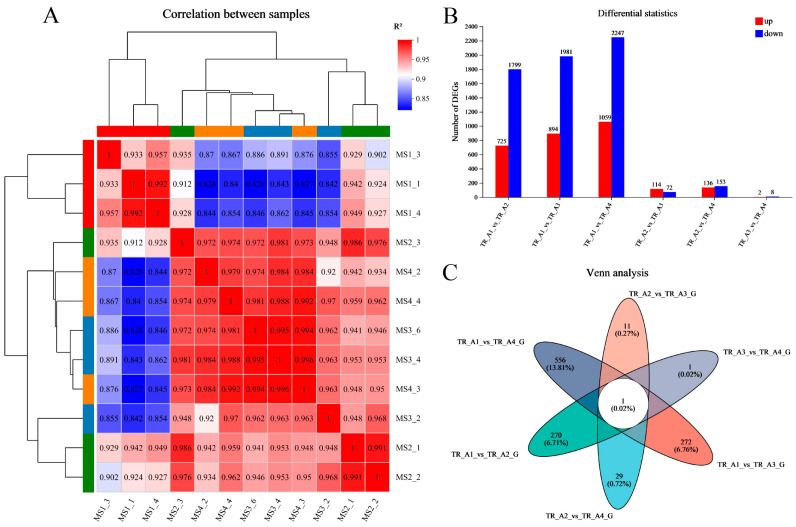
Transcriptomic data analysis. (**A**) The correlation analysis between biological replicate samples; (**B**) the differential gene comparisons between four groups; (**C**) the Venn analysis of pairwise comparison group.

**Figure 4 ijms-25-01954-f004:**
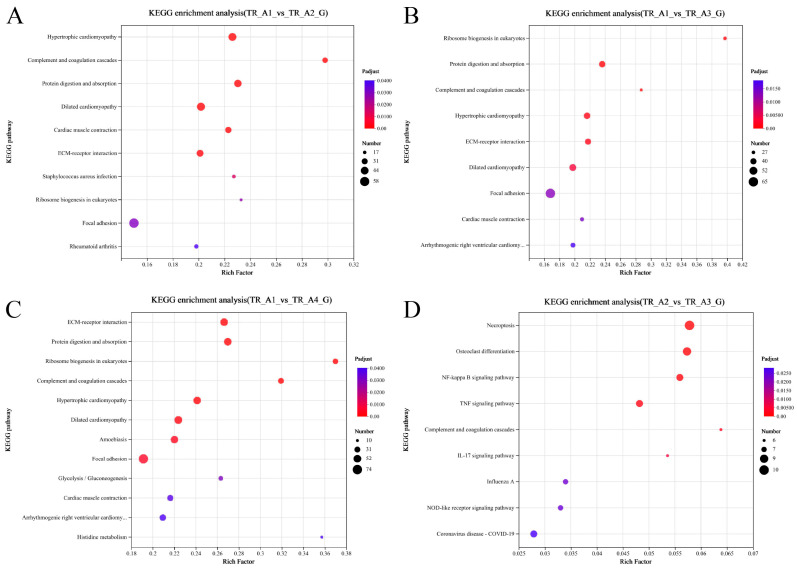
Scatterplot of the KEGG pathway enriched by differentially expressed genes. (**A**) Scatterplot of the KEGG pathway enriched by differentially expressed genes from the group of TR_A1_vs_TR_A2; (**B**) scatterplot of the KEGG pathway enriched by differentially expressed genes from the group of TR_A1_vs_TR_A3; (**C**) scatterplot of the KEGG pathway enriched by differentially expressed genes from the group of TR_A1_vs_TR_A4; (**D**) scatterplot of the KEGG pathway enriched by differentially expressed genes from the group of TR_A2_vs_TR_A3.

**Figure 5 ijms-25-01954-f005:**
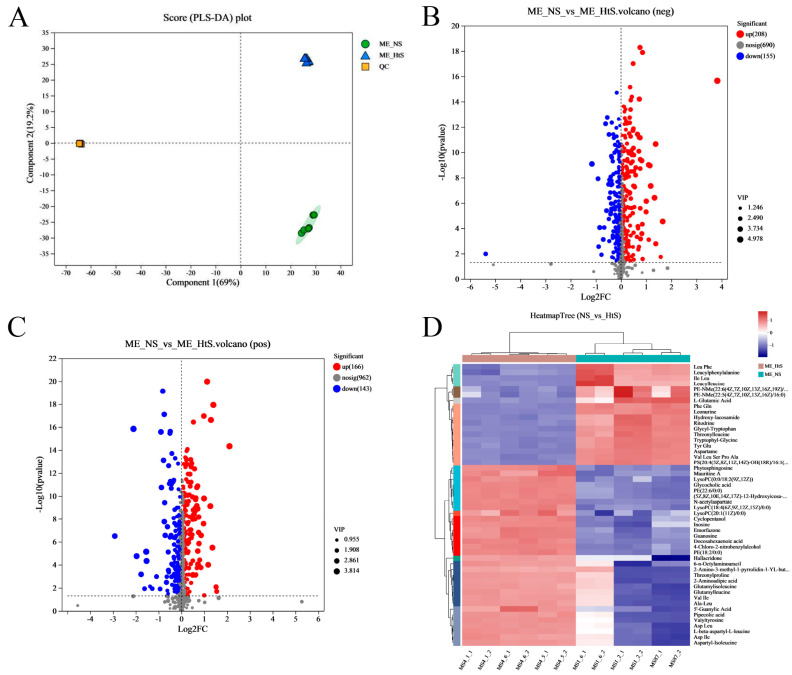
Metabonome data analysis. (**A**) The PLS−DA analysis; (**B**) the volcano plot of DMs (negative ion, neg); (**C**) the volcano plot of DMs (positive ion, pos); (**D**) the cluster and heatmap analysis result of DMs (top 50), the full name of some abbreviated metabolites, including PE−NMe− (22: 6 (4Z, 7Z, 10Z, 13Z, 16Z, 19Z)/ 18: 0), PS (20: 4 (5Z, 8Z, 11Z, 14Z)−OH (18R)/ 16: 1 (9Z)), (5Z, 8Z, 10E, 14Z, 17Z)−12−Hydroxyicosa−5, 8, 10, 14, 17−pentaenoylcarnitine, and 2−Amino−3−methyl−1−pyrrolidin−1−YL−butan−1−one.

**Figure 6 ijms-25-01954-f006:**
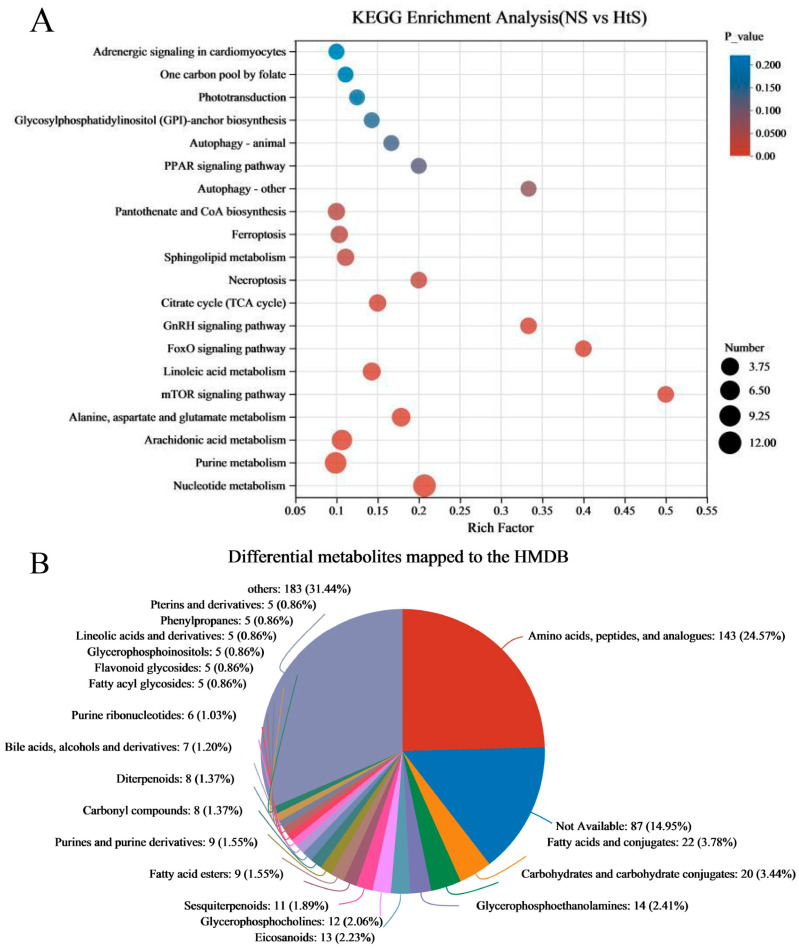
KEGG enrichment analysis and mapping classification of DMs. (**A**) Scatterplot of the KEGG pathway enriched by DMs; (**B**) the DMs mapped to the Human Metabolome Database (HMDB).

**Figure 7 ijms-25-01954-f007:**
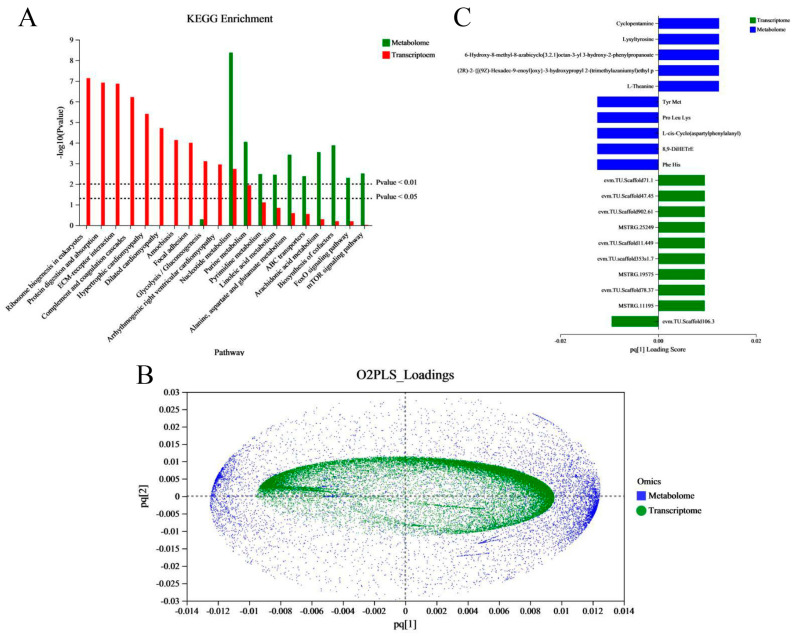
Correlation analysis between metabolites and transcripts. (**A**) Histogram of the KEGG pathways enriched by DEGs and DMs. The abscissa represents the pathway name, and the ordinate represents the enrichment significance (*p*-value) of the pathway affected. (**B**) Assessment of the intrinsic correlation between the metabolites and transcripts by the two-way orthogonal partial least squares (O2PLS) method. The abscissa and ordinate represent the combined loading value, each gene/metabolite has a relative coordinate point in “*pq* [1]” and “*pq* [2]”, “*p*” represents the loading value of the gene, and “*q*” represents the loading value of the metabolite. (**C**) The top10 DEGs and DMs loading histogram. The abscissa represents the combined loading value “*pq* [1]”, and the ordinate represents the DEGs/ DMs. (Notes, “*pq* [1]” and “*pq* [2]” only represent horizontal and vertical coordinates).

**Figure 8 ijms-25-01954-f008:**
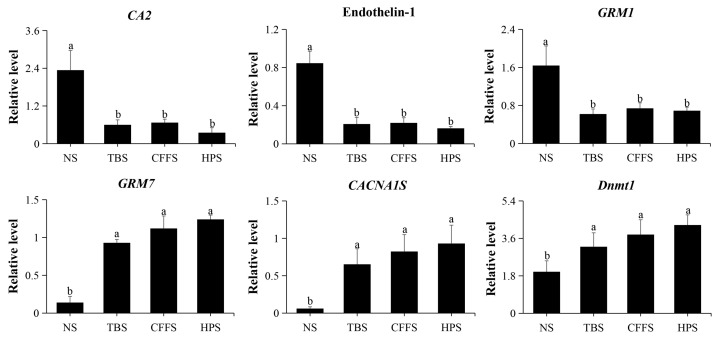
qRT-PCR validation of the relative expression levels of 6 DEGs. The abscissa represents the different stages (NS: neurula stage; TBS: tail bud stage; CFFS: caudal fin fold stage; HPS: heart pulsation stage), and the ordinate represents the relative expression level of gene. Different letters indicate values with significant difference (*p* < 0.05).

**Table 1 ijms-25-01954-t001:** The sequencing statistics for 12 RNA libraries of *Larimichthys polyactis*.

Name of Group	Sample Name	Raw Reads	Raw Bases	Clean Reads	Clean Bases	Q30 (%)	GC Content (%)	Mapped Reads	Mapped Ratio (%)
TR_A1	MS1_1	46,560,464	7,030,630,064	45,613,948	6,659,981,695	93.35	47.97	35,026,369	76.79
	MS1_3	46,079,848	6,958,057,048	45,478,904	6,718,090,478	93.96	48.55	38,418,011	84.47
	MS1_4	45,763,860	6,910,342,860	44,868,434	6,486,429,872	94.16	47.83	35,457,605	79.03
TR_A2	MS2_1	43,973,876	6,640,055,276	43,310,308	6,441,264,366	93.97	49.11	37,017,865	85.47
	MS2_2	45,550,584	6,878,138,184	44,926,238	6,662,511,904	93.91	48.63	37,569,943	83.63
	MS2_3	43,448,516	6,560,725,916	42,889,492	6,381,334,581	93.84	49.51	36,411,339	84.9
TR_A3	MS3_2	44,253,422	6,682,266,722	43,604,360	6,443,996,202	94.07	47.73	36,374,553	83.42
	MS3_4	42,974,092	6,489,087,892	42,461,328	6,280,080,349	94.12	49.46	36,352,605	85.61
	MS3_6	49,807,428	7,520,921,628	49,213,898	7,316,291,487	93.65	49.4	42,158,826	85.66
TR_A4	MS4_2	48,560,896	7,332,695,296	48,013,414	7,121,588,700	94.09	49.91	41,594,404	86.63
	MS4_3	51,011,010	7,702,662,510	50,454,496	7,500,276,460	94.14	49.08	43,603,271	86.42
	MS4_4	50,702,630	7,656,097,130	50,055,884	7,424,923,409	93.93	48.69	43,144,247	86.19

**Table 2 ijms-25-01954-t002:** The differential expression level statistics of 37 DEGs associated with the auditory system.

Gene	A1_vs_A2	A1_vs_A3	A1_vs_A4	A2_vs_A3	A2_vs_A4	A3_vs_A4	Sequence_ID
Carbonic anhydrase 2	yes|up	yes|up	yes|up	no|up	no|up	no|up	evm.TU.Scaffold208.24
Integrin α-6	yes|up	yes|up	yes|up	no|down	no|down	no|up	evm.TU.Scaffold254.286
Integrin α-5	yes|up	yes|up	yes|up	no|up	no|up	no|up	evm.TU.Scaffold804.200
Integrin α-3 CD49 antigen-like family member C	yes|up	no|up	yes|up	no|down	no|down	no|up	evm.TU.Scaffold1029.76
Otopetrin-2	yes|up	yes|up	yes|up	no|up	no|up	no|up	evm.TU.Scaffold84.30
	yes|up	yes|up	yes|up	no|up	no|up	no|up	evm.TU.Scaffold114.78
	yes|up	yes|up	yes|up	no|up	no|up	no|up	evm.TU.Scaffold114.79
Neutrophil cytosol factor 2	yes|up	yes|up	yes|up	no|up	no|up	no|up	evm.TU.Scaffold620.51
NADPH oxidase 1	yes|up	yes|up	yes|up	no|up	no|up	no|down	evm.TU.Scaffold476.222
Transmembrane protease serine	no|up	yes|up	yes|up	no|up	no|up	no|up	evm.TU.Scaffold238.12
Metabotropic glutamate receptor-1	yes|up	yes|up	yes|up	no|down	no|up	no|up	evm.TU.Scaffold169.511
Neuropilin-2	yes|up	yes|up	yes|up	no|up	no|up	no|up	evm.TU.Scaffold254.538
Integrin β-1	no|up	yes|up	yes|up	no|up	no|up	no|up	evm.TU.Scaffold620.12
Endothelin-1	yes|up	yes|up	yes|up	no|up	no|up	no|up	evm.TU.scaffold325s1.3
*Cntnap2*	yes|up	yes|up	yes|up	no|down	no|down	no|up	evm.TU.Scaffold123.15
	no|up	yes|up	yes|up	no|up	no|up	no|down	evm.TU.scaffold125s1.6
*NUFIP1*	yes|up	yes|up	yes|up	no|up	no|up	no|down	evm.TU.Scaffold133.51
Ephrin type-A receptor 7	yes|up	yes|up	yes|up	no|up	no|up	no|up	evm.TU.Scaffold169.12
*USH2A*	no|up	yes|up	yes|up	no|up	no|up	no|down	evm.TU.Scaffold69.380
NADPH oxidase 4	no|up	no|up	yes|up	no|up	no|up	no|up	evm.TU.Scaffold69.544
*Pax2*	no|up	no|up	yes|up	no|up	no|up	no|up	evm.TU.Scaffold63.125
*Lmx1a*	no|up	no|up	yes|up	no|up	no|up	no|up	evm.TU.Scaffold114.16
Carbonic anhydrase 4	yes|down	yes|down	yes|down	no|up	no|down	no|down	evm.TU.Scaffold1817.320
Collagen α-1(XI) chain	yes|down	yes|down	yes|down	no|down	no|down	no|down	evm.TU.Scaffold789.161
	yes|down	yes|down	yes|down	no|down	no|down	no|down	evm.TU.Scaffold620.61
Voltage-dependent L-type calcium channel subunit α-1S	yes|down	yes|down	yes|down	no|down	no|down	no|down	evm.TU.Scaffold411.4
Excitatory amino acid transporter 3	no|down	no|down	yes|down	no|down	no|down	no|down	evm.TU.Scaffold630.83
Fatty acid synthase	yes|down	yes|down	yes|down	no|down	no|down	no|up	evm.TU.scaffold700s1.8
Otolith Matrix Protein-1	no|down	yes|down	yes|down	no|down	no|down	no|up	evm.TU.Scaffold1603.13
α-Tectorin	yes|down	yes|down	yes|down	no|up	no|down	no|down	evm.TU.Scaffold195.21
	yes|down	yes|down	yes|down	no|down	no|down	no|down	evm.TU.Scaffold611.1
NADPH oxidase organizer 1	yes|down	no|down	yes|down	yes|up	no|up	no|down	evm.TU.Scaffold1029.136
Carbonic anhydrase 6	yes|down	yes|down	yes|down	no|up	no|up	no|down	evm.TU.Scaffold293.98
Gamma-aminobutyric acid receptor subunit β-3	yes|down	yes|down	yes|down	no|down	no|down	no|down	evm.TU.Scaffold158.122
Spectrin β chain, non-erythrocytic 1	yes|down	yes|down	yes|down	no|up	no|up	no|down	evm.TU.Scaffold195.103
Metabotropic glutamate receptor 7	yes|down	yes|down	yes|down	no|down	no|down	no|down	evm.TU.Scaffold2937.38
Transient receptor potential cation channel subfamily M member 2	no|down	yes|down	yes|down	no|down	no|down	no|down	evm.TU.Scaffold254.405
Ephrin type-B receptor 1-B	yes|down	no|down	yes|down	no|up	no|down	no|down	evm.TU.Scaffold165.282
	no|down	no|down	yes|down	no|down	no|down	no|down	evm.TU.Scaffold1030.19
Osteonectin	yes|down	yes|down	yes|down	no|down	no|down	no|down	evm.TU.Scaffold78.49
Neuroserpin	yes|down	no|down	yes|down	no|down	no|down	no|down	evm.TU.scaffold188s1.1
	yes|down	yes|down	yes|down	no|up	no|down	no|down	evm.TU.Scaffold857.38
Homeobox protein SIX1	yes|down	yes|down	yes|down	no|up	no|down	no|down	evm.TU.Scaffold1603.234
DNA (cytosine-5)-methyltransferase 1	yes|down	yes|down	yes|down	no|up	no|down	no|down	evm.TU.Scaffold282.335

“yes” and “no” represent significance of differential expression level; “up” and “down” represent the types of differential expression.

**Table 3 ijms-25-01954-t003:** The pathways co-annotated by DEGs (associated with the auditory system) and DMs.

Gene	Sequence_ID	Regulate Metabolite	Common Annotated Pathway	Pathway ID
*CA2*	evm.TU.Scaffold208.24	L-Glutamic Acid	Nitrogen metabolism	lco00910
*CA4*	evm.TU.Scaffold1817.320	L-Glutamic Acid	Nitrogen metabolism	lco00910
*CA6*	evm.TU.Scaffold293.98	L-Glutamic Acid	Nitrogen metabolism	lco00910
GABA A receptor	evm.TU.Scaffold158.122	N-Acetyl-1-aspartylglutamic acid; p-Octopamine; L-glutamic Acid	Neuroactive ligand–receptor interaction	lco04080
*GRM7*	evm.TU.Scaffold2937.38	N-Acetyl-1-aspartylglutamic acid; P-Octopamine; L-Glutamic Acid	Neuroactive ligand–receptor interaction	lco04080
*CACNA1S*	evm.TU.Scaffold411.4	PA(8:0/8:0); PA(8:0/10:0); Arachidonic acid	GnRH signaling pathway	lco04912
		Arachidonic acid	Vascular smooth muscle contraction	lco04270
		Isoproterenol	Adrenergic signaling in cardiomyocytes	lco04261
Endothelin-1	evm.TU.scaffold325s1.3	Arachidonic acid	Vascular smooth muscle contraction	lco04270
		N-Acetyl-1-aspartylglutamic acid; P-Octopamine; L-Glutamic Acid	Neuroactive ligand–receptor interaction	lco04080
*GRM1*	evm.TU.Scaffold169.511	Adenosine 5′-Monophosphate; L-Glutamic Acid	FoxO signaling pathway	lco04068
		N-Acetyl-1-aspartylglutamic acid; P-Octopamine; L-Glutamic Acid	Neuroactive ligand–receptor interaction	lco04080
		L-Glutamic Acid	Gap junction	lco04540
*Dnmt1*	evm.TU.Scaffold282.335	S-Adenosylhomocysteine; Methionine Sulfoxide; L-Cystathionine	Cysteine and methionine metabolism	lco00270

## Data Availability

The raw data of transcriptome have been uploaded to NCBI (PRJNA1012845), and other data that support the findings of this study are available from the corresponding author upon reasonable request.

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
