# Peer review of "Metabonomics and Transcriptomics Analyses Reveal the Development Process of the Auditory System in the Embryonic Development Period of the Small Yellow Croaker under Background Noise"

_ijms, 2024, doi:10.3390/ijms25041954_

Round 1

Reviewer 1 Report

Comments and Suggestions for Authors

The draft "Metabonomics and transcriptomics analysis reveal the development process of auditory system in embryonic development period of small yellow croaker under background noise" presented here is impressive, innovative and consistent. Demonstrating that noise pollution is an important problem in both natural and aquaculture environments, which needs attention. Transcriptomes of small yellow croaker made here were secured by qPCR. They through transcriptome and metabolome analysis to reveal the development process of auditory system in embryonic development period during background noise. Demonstrating that the formation process of otoliths includes occurring and calcification, suggested that the neurula stage was the importante period of otolith formation and inner ear development. It is impressive because it uses biotools to explain a complex physiological phenomenon, which can interfere with the aquaculture production process, as well as several other anthropic actions. Therefore, I encourage acceptance of the draft as is.

Author Response

Dear Reviewer,

Thank you very much for your time and dedication in reviewing our manuscript. We are very grateful for your kind appraisal. And we are honored to have had the opportunity to receive feedback from someone with your level of expertise.

Reviewer 2 Report

Comments and Suggestions for Authors

This study investigates the impact of underwater noise pollution on the embryonic development of the small yellow croaker. While previous research has primarily focused on the effects of noise on adult fishes, this study addresses the gap by analyzing embryo-larval samples at key stages of otic vesicle development. It is an interesting work, well written and showing new knowledge. The findings suggest that the auditory system development in small yellow croaker begins at least in the neurula stage, with potential implications for the impact of underwater noise pollution on early-stage embryo-larval development.

Main concerns:

1.- The absence of a control group, i.e., no noise (quite) group.

2.- The experimental design must be explained carefully in Methods section. It is not enough to mention: in this research, the embryo-larval samples were collected based on various post-fertilization time points. 

3.- Please describe the rationale of qPCR control and comparisons. Why only 1 gene was included as housekeeping.

Comments:

- Define acronyms before use: DEG, DM, HPF

Author Response

Dear Reviewer,

Thank you very much for your time and dedication in reviewing our manuscript. We are very grateful for your kind appraisal and advice.

Point 1.- The absence of a control group, i.e., no noise (quite) group.

Responds 1: Initially, we wanted to set the quiet group, and the best method was going to collect the samples of embryonic larvae in the quiet sea area. However, it is difficult to obtain the embryonic larvae of wild small yellow croaker in the sea. On the other hand, the artificial breeding process of small yellow croaker is completed indoors, and we find that there is a problem of underwater noise pollution in the breeding tank with the oxygen valve open, but the embryo development require continuous oxygen. Therefore, it is difficult to set up a no noise group under the conditions we have. In the end, we decide to go first to study the development process of auditory system in embryonic development period of small yellow croaker under aquaculture environment. In future studies, we will try to set a more contrasting group.

Point 2.- The experimental design must be explained carefully in Methods section. It is not enough to mention: in this research, the embryo-larval samples were collected based on various post-fertilization time points. 

Responds 2: Sorry for the negligence, the samples from four developmental stages that is selected based on the research report of Zhan et al. (2016) and our studies. We had added the information in Methods section, see line 533-552.

The Reference:

Zhan, W.; Lou, B.; Chen, R.Y.; et al. Observation on embryonic development and morphological characteristics of Larimichthys polyactis. Oceanologia Et Limnologia Sinica. 2016, 5, 7. https://doi.org/10.11693/hyhz20160500114.

Point 3.- Please describe the rationale of qPCR control and comparisons. Why only 1 gene was included as housekeeping.

Responds 3: Thank you very much for the suggestion, this information was added in revised manuscript, see line 650-671. We had used another housekeeping gene (glyceraldehyde-3-phosphate dehydrogenase, GAPDH) to use for secondary validation. The results were consistent with the first validation (see line 309-311, Supplementary results). In this study, we chose β-actin as a housekeeping gene that is because we referred some of the reported studies which was using small yellow croaker as subjects [1-5], they suggested that β-actin is a stable reference gene in the studies of small yellow croaker, meanwhile, β-actin is also a reference gene that we have used more in previous studies. 

The Reference:

[1] Zhang, X.; Zhou, J.; Xu, W.; et al. Transcriptomic and Behavioral Studies of Small Yellow Croaker (Larimichthys polyactis) in Response to Noise Exposure. Animals 2022, 12, 2061. https://doi.org/10.3390/ani12162061

[2] Xue, H.; Mingming, H.; Wei, Z.; et al. Mixture effects of imidacloprid and difenconazole on enzymatic activity and gene expression in small yellow croakers (Larimichthys polyactis). Chemosphere 2022, 306, 135551. https://doi.org/10.1016/j.chemosphere.2022.135551

[3] Ran, X.; Zuting, G.; Li B.Z.; et al. Neuropeptide FF-related gene in fish (Larimichthys polyactis): identification, characterization, and potential anti-inflammatory function. Molecular Biology Reports 2022, 49, 6385-6394. https://doi.org/10.1007/s11033-022-07447-5

[4] Feng, L.; Tianle, Z.; Yu, H.; et al. Integration of transcriptome and proteome analyses reveals the regulation mechanisms of Larimichthys polyactis liver exposed to heat stress. Fish & Shellfish Immunology 2023, 135, 108704. https://doi.org/10.1016/j.fsi.2023.108704

[5] Zhang, X.; Tang, X.; Xu, J.; et al. Transcriptome analysis reveals dysfunction of the endoplasmic reticulum protein processing in the sonic muscle of small yellow croaker (Larimichthys polyactis) following noise exposure. Marine Environmental Research 2024, 194, 106299. https://doi.org/10.1016/j.marenvres.2023.106299

Comments:

- Define acronyms before use: DEG, DM, HPF

Responds 4: Thanks for your suggestion. We add these definitions in Abstract section, see line 27-29. In the main text of manuscript, we had commented these words when they were appeared for the first time. If necessary, we will add a comment table in our manuscript.

Reviewer 3 Report

Comments and Suggestions for Authors

In the manuscript entitled “Metabonomics and transcriptomics analysis reveal the develop-2 ment process of auditory system in embryonic development period of small farmed yellow croaker under background noise”, the authors assessed the potential effects of underwater noise pollution on the otic vesicle development of yellow croaker through metabolomics and transcriptomics analyses. The work involves an integrated analysis with a complete bioinformatics analysis, and results are well interpreted. The authors used the terms “Metabonomics” in the title and abstract, but throughout the manuscript, “Metabolomics” was used. These terms should not be used interchangeably, the authors should uniformize the terms employed. The figures font size is really small, I suggest increasing it to improve readability. The experimental design is extremely incomplete: there is no indication of number of fish used, number of tanks, number of biological, technical and analytical replicates, physicochemical parameters, etc. When mentioning/citing software and packages, it is mandatory to include the version used. The manuscript has some serious English grammar issues; a thorough language review is extremely required.

Line 15 – I believe the authors meant “research” and not “researcher”

Line 17 – “Recently” is more correct than “In recent”

Line 536 – what were the software used for raw reads preprocessing? This should be included 

Comments on the Quality of English Language

The manuscript has some serious English grammar issues; a thorough language review is extremely required.

Author Response

Dear Reviewer,

Thank you very much for your time and dedication in reviewing our manuscript. We are very grateful for your kind appraisal and advice.

We had using “Metabonomics” and “Metabonome” to replace “Metabolomics” and “Metabolome”.

About the figures, we were uploading the original images, and they could be clearly enlarged. Possibly, some were composed of multiple pictures, which will look smaller, or the problem of manuscript format. If necessary, we are willing to make appropriate changes to the figures.

Sorry for our negligence, we had added the procedure and instructions of experimental design in our manuscript, see line 533-551. Meanwhile, we supplemented the parameter description, including number of parent fishes, number of collected embryonic larvae, egg diameter, number of bucket, volume of bucket, etc (see line 541-550, line 565-566, line 595-596), and we supplemented the version number of software and packages (see 4.2, 4.4, 4.5, 4.7 and 4.8 of “Materials and Methods”).

Point 1. Line 15 – I believe the authors meant “research” and not “researcher”

Responds 1: Sorry for the negligence, we had changed this word in line 16.

Point 2. Line 17 – “Recently” is more correct than “In recent”.

Responds 2: Thanks for your suggestion. We had replaced this word in line 19.

Point 3. Line 536 – what were the software used for raw reads preprocessing? This should be included. 

Responds 3: Thank you very much for the suggestion. We had added this software information in line 560- 561.

Point 4. The manuscript has some serious English grammar issues; a thorough language review is extremely required.

Responds 4: Thank you very much for your suggestion. We had used the English Editing Service of MDPI (Edit text number: english-edited-76374) to improve the quality of our manuscript.

Round 2

Reviewer 2 Report

Comments and Suggestions for Authors

I appreciate the consideration of the comments. The manuscript has been significantly improved as a result, and I believe it is now suitable for acceptance.

Reviewer 3 Report

Comments and Suggestions for Authors

The authors have addressed my main concerns and the manuscript is well improve. I recommend acceptance as it is